# Gaelic Football Match-Play: Performance Attenuation and Timeline of Recovery

**DOI:** 10.3390/sports8120166

**Published:** 2020-12-17

**Authors:** Lorcan S. Daly, Ciarán Ó Catháin, David T. Kelly

**Affiliations:** 1Department of Sport and Health Sciences, Athlone Institute of Technology, N37 HD68 Athlone, Ireland; ciaranocathain@ait.ie (C.Ó.C.); davidkelly@ait.ie (D.T.K.); 2SHE Research Group, Athlone Institute of Technology, N37 HD68 Athlone, Ireland

**Keywords:** Gaelic football, fatigue, post match, muscle damage, countermovement jump, recovery, post-match fatigue, match-play, GPS, monitoring

## Abstract

This study investigated acute changes in markers of fatigue and performance attenuation during and following a competitive senior club-level Gaelic football match. Forty-one players were tested immediately pre-match, at half-time, full-time, 24 h post-match and 48 h post-match. Creatine kinase, drop jump height and contact-time, reactive strength index, countermovement jump height and perceptual responses were assessed at the aforementioned time-points. 18 Hz global positioning system devices were used to record players in-game workload measures. Compared to pre-match, perceptual responses (−27.6%) and countermovement jump height (−3.9%) were significantly reduced at full-time (*p* < 0.05). Drop jump height (−8.8%), perceptual responses (−27.6%), reactive strength index (−15.6%) and countermovement jump height (−8.6%) were significantly lower 24 h post-match (*p* < 0.05). Pre-match creatine kinase was significantly increased (+16.2% to +159.9%) when compared to all other time-points (*p* < 0.05). Total distance, total accelerations, total sprints, sprint distance and average heart rate were all correlated to changes in perceptual responses (r = 0.34 to 0.56, *p* < 0.05). Additionally, maximum speed achieved (r = 0.34) and sprint distance (r = 0.31) were significantly related to countermovement jump changes (*p* < 0.05), while impacts (r = 0.36) were correlated to creatine kinase increases (*p* < 0.05). These results demonstrate that Gaelic football match-play elicits substantial neuromuscular, biochemical and perceptual disturbances.

## 1. Introduction

Gaelic football is a field-based team sport native to Ireland which is contested on a grass pitch with two teams of 15 players [1]. It is an intermittent invasion field sport, whereby intensive anaerobic efforts occur in a cyclical nature behind a background of light aerobic activity [1]. The game is fast paced, with frequent turnovers of possession, necessitating players to have well developed components of fitness and technical skills [1,2]. During competitive matches, players typically cover large total distances (5 km to 11 km) with variation in workloads and physical demands depending on the playing level, player position, tactics and many other factors [3,4]. Elite-level players are reported to cover 1563 ± 605 m of high-speed running distance (≥4.7 m·s^−1^), 524 ± 190 m of very high speed running distance (≥6.1 m·s^−1^) and perform 166 ± 41 m of accelerations [2].

During Gaelic football match-play, workloads following intensive periods of play are temporarily impeded due to performance attenuation, while work rates progressively decline over the course of a game [4]. In elite-level Gaelic football, a significant progressive reduction in total and high speed running distance (≥5.5 m·s^−1^) is reported across the final three quarters when compared with the first quarter [3]. While such research is beneficial in quantifying the workloads which players undertake during a game, there is a lack of information surrounding players’ responses during the post-match recovery timeline and their relationships with the in-game workloads.

The demands that team sport players are exposed to during match-play likely invokes considerable immediate and prolonged decrements in contractile function (peripheral fatigue) and the capacity of the central nervous system to activate muscles (central fatigue) [5,6]. Fatigue following Gaelic football match-play may be attributed to a combination of metabolite accumulation, impaired nervous system control, energy system depletion and a functional decline of the muscle fibers, contractile mechanisms similar to other team sports [7,8,9,10]. Research from team sports with comparable on-field demands to Gaelic football suggests players are likely to endure substantial perturbations in neuromuscular function, muscle damage and increased perceptions of tiredness and muscle soreness [6,11,12]. Previous work evaluating post-match fatigue and muscle damage across a range of team sports has used a variety of methods, including neuromuscular, biochemical, endocrine and perceptual markers, to assess players’ responses [8,9,13].

When evaluating responses to match-play, neuromuscular markers, such as countermovement jump (CMJ) and drop jump (DJ), are particularly useful because they are relatively non-fatiguing, fast to complete and utilize the stretch shortening cycle (SSC) of the lower body musculature, allowing an accurate depiction of an athletes’ neuromuscular condition [14]. A combination of neuromuscular and biochemical markers is reported to be highly reflective of subcellular and cellular muscular disturbances following intense eccentric actions and physical contact during games and may provide valuable indications of the degree of muscle damage following Gaelic football match-play [15,16]. Previous discrepancies observed between objective neuromuscular and biochemical markers and subjective perceptual responses (PR) have provided rationale for the inclusion of a perceptual questionnaire to extensively illustrate the early warning signs of fatigue, overtraining and post-match recovery kinetics [9,17]. Furthermore, in-game workload measures, such as high speed running distance and sprint frequency, have also been reported to be highly sensitive monitoring tools when characterizing post-match neuromuscular and biochemical disturbances [18,19]. Subsequently, in order to accurately evaluate the impact of Gaelic football match-play, a multifactorial testing approach may be required to comprehensively assess players’ responses and post-match recovery profiles [10].

Reliance on data from other sports is currently necessary for the determination of training load, recovery practices and performance profiling of Gaelic football players, providing sub-optimal guidance to coaches and players within the sport [12,15,20,21]. Different team sports are reported to have unique post-match recovery kinetics due to large variations in competitive demands [10]. Additionally, the amateur status of Gaelic football provides environmental factors, such as work and study, which are important to consider during post-match recovery [22]. Consequently, effectively preparing for subsequent training or competition necessitates the monitoring of Gaelic football players during the post-match recovery timeline. Such knowledge is vital for injury prevention and nutritional strategies, tactical player rotation and the optimal management and periodization of training loads [10,23,24,25]. Therefore, this research aims to investigate the changes in markers of fatigue and performance attenuation during and following competitive Gaelic football match-play. A secondary aim of this study is to examine the relationship between in-game workload measures and changes in the markers of fatigue and performance attenuation.

## 2. Materials and Methods

### 2.1. Study Design

A familiarization session was used to explain the testing procedures to the participants 1 week before testing. Participants’ neuromuscular, perceptual and biochemical markers were tested pre-match, at half-time, full-time, 24 h post-match and 48 h post-match (Figure 1). The participants’ in-game workloads were recorded during a competitive match. Pre-match testing was assessed after the participants had completed their regular team warm up. Participants were asked to refrain from any form of alcohol consumption or strenuous activity in the 24 h prior to and during the match-day and post-match testing days. Additionally, participants were instructed to arrive hydrated and well rested, and to avoid caffeine three hours before the testing sessions.

### 2.2. Participants

Forty-one physically active and healthy male participants (mean ± SD, age: 23.3 ± 4.2 years; height: 178.3 ± 7.91 cm; body mass: 80.64 ± 9.47 kg, sum of 7 skinfolds: 81.3 ± 28.0, percentage body fat: 14.3 ± 5.2) between 18–32 years of age currently playing senior club-level Gaelic football volunteered to participate in the study. Each participant had a minimum of 2 years of resistance training experience and 3 years of experience playing adult-level Gaelic football. Participants in the present study were recruited from a convenience-based sample from 5 clubs in the local region playing Gaelic football at senior level. Throughout the competitive season, participants trained an average of 3 days each week, with a mixture of gym and field-based training sessions, and were involved in Gaelic football matches predominantly at weekends. Participants were omitted from the study if they had suffered any lower body musculoskeletal injury in the past 2 months or failed to pass a Physical Activity Readiness Questionnaire (PAR-Q) assessment. Informed consent was acquired from all participants in accordance with the Athlone Institute of Technology (AIT) guidelines. The AIT Research Ethics Committee granted ethical approval for this research (code 20180501).

### 2.3. Anthropometrics and Body Composition

Body mass and height were measured to the nearest 0.1 kg and 0.1 cm, respectively, using a portable stadiometer and scales (Seca 707 Balance Scales, GmbH, Hamburg, Germany). Skinfold thickness was measured at 7 anatomical sites (abdomen, midaxillary, biceps, chest, thigh, triceps, and subscapular) using handheld Harpenden Skinfold callipers (Baty International Ltd., Sussex, UK). While the investigator was not accredited with the International Society for the Advancement of Kinanthropometry (ISAK), pilot testing was used to verify the accuracy of the anthropometrical measurements performed [26,27]. When intra-rater reliability was assessed for skinfolds, the technical error of measurement of 4 repeated trials was lower than 5%, which is in line with recommendations [26]. Triple measurements for skinfolds were obtained to the nearest 0.2 mm on the right side of the body using the ISAK protocols [27]. If the difference was greater than 2 mm for any measurement, a fourth measurement was taken. To calculate participants’ body fat, the equation of Withers et al. [28] was used (% body fat = 495/(1.0988 − 0.0004 × [sum of 7 skinfolds]) − 450). This equation has been commonly used in the case of team sport athletes [29] and has been reported to have the lowest bias and highest relationship and agreement with dual-energy absorptiometry (DEXA) referenced values when compared with a number of other equations [28,30].

### 2.4. Sampling of Plasma Creatine Kinase (CK)

Concentrations of participants’ creatine kinase (CK) were obtained from capillary blood, collected via a finger-prick (32 μL) on the middle or index finger [31] and analysed by means of a colorimetric assay process (Reflotron Plus reader, Roche Diagnostics, Mannheim, Germany). Pre-match, half-time and full-time samples were taken at the club grounds, while the 24 h and 48 h post-match samples were taken in the sport science laboratory at AIT. All samples were analysed immediately upon collection. Calibration, testing and storage of strips were undertaken according to the manufacturers’ specifications [32].

### 2.5. Perceptual Responses (PR)

A 5-question Likert scale questionnaire was chosen to assess the participants’ perceptual status [33,34,35]. The questionnaire involved a graded Likert system of answers, with 5 questions on fatigue, sleep quality, general muscle soreness, mood and levels of stress. The readings were rated 1–5, with increments of 0.5. Overall perceptual wellbeing was determined through the addition of the five scores. All of the participants completed the questionnaire in private behind a folding screen, either at the club grounds or in the sport science laboratory at AIT, so as to avoid external influence from coaches and/or other players [9].

### 2.6. Countermovement Jump (CMJ) and Drop Jump (DJ)

The CMJ and DJ were performed using the Optojump (Optojump, Microgate, Bolzano, Italy) photoelectric optical measurement device. The subjects stood between the Optojump bars on a standardised surface, hands placed on their hips, and jumped vertically to a maximum height [36]. A 30 cm high plastic step was used throughout all DJ testing. Participants stood on this step with their hands placed on their hips and stepped off with their dominant foot first (both feet landing on the floor simultaneously) before immediately jumping vertically as high as possible. Participants were instructed to minimise ground contact time. Three CMJ and three DJ trials were performed, with 1 min of rest between attempted jumps. The reactive strength indexes (RSI) of the players were calculated by dividing the height jumped (m) in the DJ by the ground contact time (s) [36]. For all CMJ and DJ measures, the average of three attempts was used in all match-day and post-match testing, as this has been reported to be more sensitive to players’ recovery status [14].

### 2.7. Global Positioning System (GPS) Monitoring

Participants’ in-game workloads were recorded using 18 Hz GPS units (Apex, STATSports, Newry, UK). A heart rate (HR) telemetry system (Polar Vantage NV Polar, Port Washington, NY, USA) was placed around the chest to collect HR data simultaneously. The participants were fitted with an appropriately sized custom vest that held in place the GPS receiver. Approximately 15 min before the game’s throw in, the GPS unit was powered on and then inserted into a padded slot towards the top of the rear section of the vest, sitting between the scapulae in the upper thoracic spinal region. The HR telemetry strap was placed through the custom-made slits in the front of the vest at the level of the chest. The GPS unit was switched off immediately following completion of the game. Exact times and durations of throw ins, stoppages, additional time or any tactical substitutions or sending offs were recorded manually by the researcher. Players were only included in the study if they completed ≥60 min of match-play [3].

### 2.8. Statistical Analysis

Descriptive statistics using mean and standard deviation (±SD) were calculated for all anthropometrics (age, body mass, height and sum of 7 skinfolds) and in-game workload measures (max speed (m·s^−1^), distance (m), accelerations, sprints (≥5.5 m·s^−1^), total impacts (>2 g in 0.1 s), explosive distance (m), average HR (beats.m^−1^)). The assumption of normality was assessed using the Shapiro–Wilk test. All data passed the assumption of normality, according to the Shapiro–Wilk test (*p* > 0.05). Multiple repeated measure ANOVAs with Bonferroni post-hoc analysis were used to compare biochemical, perceptual and neuromuscular measures (CK, PR, DJ, RSI, CT and CMJ) across each of the time-points (pre-match, half-time, full-time, 24 h post-match and 48 h post-match). Pearson product–moment correlation coefficient analysis was used to assess the strength of the linear relationships between changes in markers of fatigue and performance attenuation and in-game workloads. Multiple repeated measure ANOVAs were used to guide the selection of two time-points with the largest number of significant disturbances to markers of performance attenuation and fatigue (CK, PR, DJ, RSI and CMJ) for Pearson product–moment correlation coefficient analysis. The Holm–Bonferroni correction was employed for multiple correlations at each time-point to reduce the risk of type I errors [37]. All data were analyzed using Statistical Package for Social Sciences (SPSS Version 26, SPSS Inc., Chicago, IL, USA). Statistical significance was set at an alpha level of *p* < 0.05.

## 3. Results

### 3.1. In-Game Workload Measures

Table 1 outlines the workload metrics of the players during the first half, second half and the full game.

### 3.2. Changes in Measurments of Fatigue and Performance Attenuation

#### 3.2.1. Drop Jump Height (DJ), Contact Time (CT) and Reactive Strength Index (RSI)

Figure 2A outlines the drop jump height results over the five assessed time-points (pre-match, half-time, full-time, 24 h and 48 h post-match). Post hoc analysis indicated that there was a significant decrease between pre-match and 24 h post-match (−8.8%, *p* < 0.001), half-time and full-time (−5.1%, *p* = 0.03), half-time and 24 h post-match (−9.9%, *p* < 0.001) and half-time and 48 h post-match (−5.2%, *p* = 0.005). A significant increase was also found between 24 h post-match and 48 h post-match (+5.7%, *p* < 0.001). No significant differences were observed when comparing DJ CT at any of the five time-points (*p* > 0.05) (Figure 2B). Figure 2C illustrates the RSI data collected over the five assessed time-points (pre-match, half-time, full-time, 24 h post-match and 48 h post-match). Post hoc analysis indicates that there was a significant decrease between pre-match and 24 h post-match (−15.6%, *p* = 0.014) and half-time and 24 h post-match (−14.1%, *p* = 0.010).

#### 3.2.2. Countermovement Jump (CMJ)

Figure 2D illustrates the CMJ height results collected over the five time-points (pre-match, half-time, full-time, 24 h post-match and 48 h post-match). Post hoc analysis indicates that there was a significant decrease between pre-match and full-time (−3.9%, *p* = 0.039), pre-match and 24 h post-match (−8.6%, *p* < 0.001). Additionally, significant decreases were found between half-time and full-time (−5.2%, *p* < 0.001), half-time and 24 h post-match (−9.9%, *p* < 0.001), half-time and 48 h post-match (−4.6%, *p* = 0.040) and 24 h post-match and 48 h post-match (+5.8%, *p* < 0.001).

#### 3.2.3. Creatine Kinase (CK)

Figure 3A illustrates the creatine kinase results over the five time-points (pre-match, half-time, full-time, 24 h and 48 h post-match). Post hoc analysis results indicate that there was a significant increase between pre-match and half-time (+16.2%, *p* < 0.001), pre-match and full-time (+43.8%, *p* < 0.001), pre-match and 24 h post-match (+159.9%, *p* < 0.001) and pre-match and 48 h post-match (+70.1%, *p* < 0.001). A significant increase was also found between half-time and full-time (+23.7%, *p* < 0.001), half-time and 24 h post-match (+123.5%, *p* < 0.001) and half-time and 48 h post-match (+46.3%, *p* < 0.001). Additionally, a significant increase was found between full-time and 24 h post-match (+80.7%, *p* < 0.001) and a significant decrease between 24 h post-match and 48 h post-match (−18.3%, *p* < 0.001).

#### 3.2.4. Perceptual Questionnaire

Figure 3B illustrates the perceptual score results over the five assessed time-points (pre-match, half-time, full-time, 24 h and 48 h post-match). Post hoc analysis indicates that there was a statistically significant decrease between pre-match and half-time (−8.7%, *p* < 0.001), pre-match and full-time (−27.6%, *p* < 0.001) and pre-match and 24 h post-match (−18.0%, *p* < 0.001). A significant decrease was also found between half-time and full-time (−20.8%, *p* < 0.001) and half-time and 24 h post-match (−10.5%, *p* < 0.001). Additionally, a significant increase was also found between full-time and 24 h post-match (+9.6%, *p* < 0.001), full-time and 48 h post-match (+22.1%, *p* < 0.001) and 24 h post-match and 48 h post-match (+12.5%, *p* < 0.001).

#### 3.2.5. Correlations between Markers of Fatigue and Performance Attenuation and In-Game Workloads

Changes in PR from pre-match to full-time were correlated to total distance (r = 0.48, *p* = 0.014), total accelerations (r = 0.34, *p* = 0.029), total sprints (r = 0.37, *p* = 0.018) and sprint distance (r = 0.56, *p* < 0.001). PR change from pre-match to 24 h post-match was correlated with total sprints (r = 0.37, *p* = 0.017) and average HR (r = 0.36, *p* = 0.022). CMJ change from pre-match to full-time correlated with max speed (r = 0.34, *p* = 0.032), while sprint distance correlated with CMJ change from pre-match to 24 h post-match (r = 0.31, *p* = 0.049). Finally, CK change from pre-match to 24 h post-match was significantly correlated with number of impacts (r = 0.31, *p* = 0.048).

## 4. Discussion

The aim of the current study was to examine changes in markers of fatigue and performance attenuation during and following a competitive senior club-level Gaelic football match. A secondary aim was to investigate the relationships between in-game workload measures and these changes in markers of fatigue and performance attenuation. To the authors knowledge, this is the first study to document changes in markers of fatigue and performance attenuation in Gaelic football match-play.

Our results suggest match-play causes significant increases in CK across all time-points in comparison to pre-match baseline values. The largest increases were recorded at 24 h (159.9%) and 48 h (70.1%) post-match compared to pre-match values, which is in agreement with previous studies examining changes in CK following soccer [15], Australian rules football [38] and rugby league [20] match-play. An elevated presence of plasma muscle proteins in the form of CK at these time-points is associated with muscle damage, a prolonged suppression in force production, muscular soreness and increased markers of inflammation in both the muscle itself and leaked into the bloodstream [16,39]. The relative increases in CK observed in the present analysis are 25.7% to 43.1% higher 24 h post-match and 24.3% to 54.9% higher 48 h post-match in comparison with professional rugby league, soccer and Australian rules football [6,40,41]. Our work suggests the number of impacts recorded (133.5 ± 122.4) during Gaelic football match-play correlates with increases in CK 24 h post-match (r = 0.34). A similar link between physical contact and elevations in CK during the post-match recovery timeline has been previously reported [42,43,44]. Physical contact in Gaelic football involves frequent shoulder charges and tackling when contesting for possession [1]. Damage to muscle tissue from the blunt force trauma of physical contact and eccentric actions during high speed running is suggested to increase cell permeation of muscle enzymes into the bloodstream and may explain the rises in post-match CK [16]. The amateur status of Gaelic football may be indicative of less developed components of fitness when compared with the professional athletes described in the above research, thus partly explaining the relatively larger increases observed in CK [20,45]. The present cohort covered on average 118 m·min^−1^ (Table 1) is similar to previous work on this population (119 m·min^−1^) [4], but lower then elite-level soccer players (126 m·min^−1^) [46] or Australian rules football players (127 m·min^−1^) [47], who undertake larger relative workloads. Interestingly, players’ pre-competition CK levels were 7.1% higher than pre-match levels for professional rugby league players, 7.8% higher than Australian rules players and 18.8% higher than soccer players [38,41,48]. This may be due to residual fatigue and muscle damage from previous training or competition, especially considering this study was conducted during the competitive season where Gaelic football players must balance busy training and match schedules alongside working life, as amateur athletes [22,35]. Elevated CK levels prior to competition have been linked with an increased risk of injury [11,16] and attenuated match performance [13]. In the present data, elevated CK prior to match-play may provide evidence of residual fatigue and suggest that players may not be sufficiently recovered prior to matches during the competitive season.

Neuromuscular decrements recorded following Gaelic football match-play are similar to those reported during the recovery timeline of other team sports [6,20,49]. CMJ was observed to decline at full-time (−3.9%) and 24 h post-match (−8.6%), while reductions in RSI (−15.6%) and DJ height (−8.8%) became evident 24 h post-match compared to pre-match, suggesting players neuromuscular capacity is at its lowest the day after a match. While there was a significant deterioration in DJ height and RSI performance 24 h post-match, there was no significant change in CT. This may suggest that neuromuscular fatigue and muscle damage 24 h post-match reduces players rate of force development during the DJ, without having a significant impact on ground reaction time. These findings echo previous reports highlighting a delayed peak in neuromuscular decrements relating to muscle damage and inflammatory responses following strenuous SSC activity [11,15], which may explain the larger neuromuscular disturbances reported 24 h post-match [11,39]. Similar impairments in neuromuscular function have been shown to negatively affect players capacity to undertake high intensity running, total distance, ball disposals and coaches ratings of performance in Australian rules [13], while compromising the performance of technical skills and high intensity running in soccer [7,50]. This information may be helpful to coaches planning training sessions in the days following a Gaelic football match, where players’ capacity to perform high intensity running or technical skills would likely be diminished.

In the current analysis significant differences in perceptual status were observed between pre-match and full-time (−27.6%), 24 h post-match (−18%) and 48 h post-match (+12.5%). Perception of fatigue, muscular soreness and overall wellbeing have been previously documented to be sensitive to changes in players neuromuscular status, similar to markers of muscle damage and neuromuscular parameters [9,33,35]. Surprisingly, the lowest PR score was recorded at full-time, while all other markers of performance attenuation presented with the largest decrement 24 h post-match. The reported differences in peak disturbance between these objective and subjective indicators of performance attenuation corresponds to previous research on team sport athletes [9,35]. Discrepancies between neuromuscular, biochemical and PR markers during the recovery timeline have been suggested to provide rationale for the inclusion of a PR questionnaire to comprehensively monitor players physical and psychological preparedness to train or compete [17,33]. In this analysis, PR displayed the greatest decrement at FT with a recovery to baseline at 48 h, suggesting players may have reduced perceptions of fatigue and muscular soreness in the presence of ongoing biochemical and neuromuscular disturbances. Therefore, it may be important to not exclusively rely on PR when evaluating players recovery profiles [9,33]. The divergence between PR and the other markers of performance attenuation highlight the multifactorial nature of fatigue and muscle damage evoked by Gaelic football match-play and implies that numerous monitoring tools may be necessary to optimally determine players readiness to compete or train.

In the present study, disturbances to PR, CK and CMJ following match-play positively correlated with total distance, total accelerations, total sprints, max speed, sprint distance and average HR (r = 0.34 to 0.56). While the reported correlations may be weak, these findings are in agreement with previous research suggesting an association between in-game workloads and post-match fatigue [18,38], with high speed running distance (≥5.5 m·s^−1^) has been reported to be an especially sensitive monitoring variable (r = 0.44 to 0.67) [18,19]. The correlations between high intensity running indices and post-match disturbances in the current analysis may be explained by the large mechanical loads and rapid transitions from eccentric to concentric actions involved in sprinting and accelerations, which have previously been associated with greater muscle damage [18,38]. In addition, the intense eccentric actions during the landing phase of sprinting movements have been hypothesised to be a key contributor to the elevation of CK throughout the post-match recovery timeline [15,19,20]. Such associations between in-game workloads and post-match fatigue have also been correlated to players’ characteristics of fitness in team sport, which could be a factor in the present analysis [51,52] and may warrant future research. Specifically, players with higher levels of conditioning are reported to undertake greater competitive workloads, while presenting with improved post-match recovery responses then their less conditioned counterparts [52,53]. Overall, these findings suggest that high intensity running indices during match-play may provide a useful indicator to subsequent perceptual, biochemical and neuromuscular disturbances. This may be particularly useful in scenarios where teams do not have access to CK or neuromuscular assessments but can instead use measurements of in-game workload to characterise post-match responses.

In summary, the results from this study suggest there are large decrements in performance and substantial multidimensional fatigue experienced by players following Gaelic football match-play. Specifically, PR, DJ, CMJ, RSI and CK presented with the greatest disturbances 24 h post-match, while PR were lowest at FT. The recovery of neuromuscular and perceptual measures was observed 48 h post-match in the presence of continued biochemical disturbance. Consequently, players’ capacity to perform intense exercise is likely to be impaired during the 48 h post-match recovery timeline. Additionally, total accelerations, max speed, total distance, total sprints, average HR and sprint distance may be used as tracking variables to characterise changes in PR, CK and CMJ following match-play. The faster trend for PR recovery may suggest coaches need an additional objective parameter to accurately determine players’ recovery profile. Considering the variation in responses observed across the utilised markers, coaches may require more than one monitoring tool to effectively establish players’ preparedness for subsequent training or competition. Understanding players’ responses to match-play may allow for a more tailored manipulation of training and recovery modalities to improve their ensuing competitive performances.

A number of limitations exist in the present study which should be considered. Firstly, a maximum of five players could be assessed per game, meaning there was variation in match conditions, opposition, match outcome and other tactical influences which may have influenced results [20]. Additionally, the final post-match testing was undertaken 48 h post-match, with all parameters not yet recovering to pre-match values. A 72 h window may have been enough to witness a return to pre-match baseline; however, most players had a subsequent training session before this time-point. The current study was also limited by its sample size and the fact that the players were only monitored around one competitive match during mid-season. Monitoring a larger sample of players at different stages of a season may yield a better description of players’ responses to match-play. In addition, while players were asked to refrain from exercise during the 48 h post-match timeline, we could not ensure players followed these recommendations. Other external factors such as physical conditioning, nutrition, work stress and recovery strategies were not controlled and may have influenced our findings [24].

## 5. Conclusions

Competitive Gaelic football match-play elicited significant disturbances to players’ neuromuscular and perceptual markers up to 24 h post-match, while full recovery of these variables occurred 48 h post-match despite a prolonged elevation in the biochemical marker CK. Performance of high intensity running and acceleration indices coincide with greater decrements to neuromuscular and perceptual status following match-play, suggesting that monitoring selected high intensity running variables during competitive games may help contribute to customized training load management during the 48 h post-match window. Further research is necessary to ascertain the influence of players’ components of fitness and performance characteristics on in-game workload and post-match recovery kinetics. Understanding how players respond to match-play and the subsequent time course of recovery will allow coaches to effectively plan ensuing training sessions and prepare for upcoming competitive matches. When making decisions regarding players’ recovery status, coaches should consider the context of long-term and short-term recovery. For example, in situations where important matches are a few days apart, neuromuscular performance may have recovered to pre-match values in the presence of biochemical disturbances. This may be sufficient in the short-term, provided full recovery is possible after these matches. However, if a high density of matches exists for an extended period without an upcoming phase of rest, a reduction of training load and implementation of recovery strategies may be necessary to avoid cumulative fatigue. This may be especially important in scenarios where coaches do not have access to biochemical assessments, and insufficient recovery may not be detected by perceptual or neuromuscular markers.

## Figures and Tables

**Figure 1 sports-08-00166-f001:**
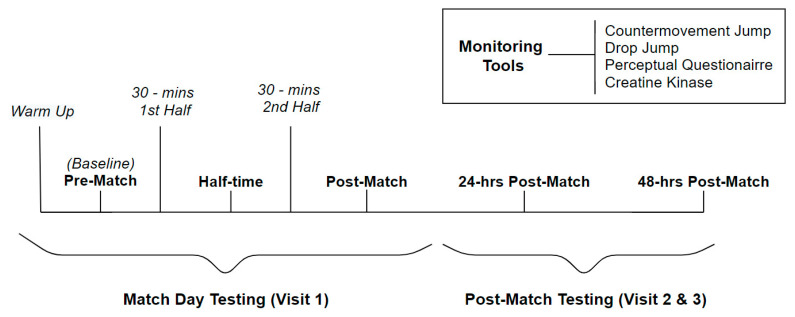
Experimental study design.

**Figure 2 sports-08-00166-f002:**
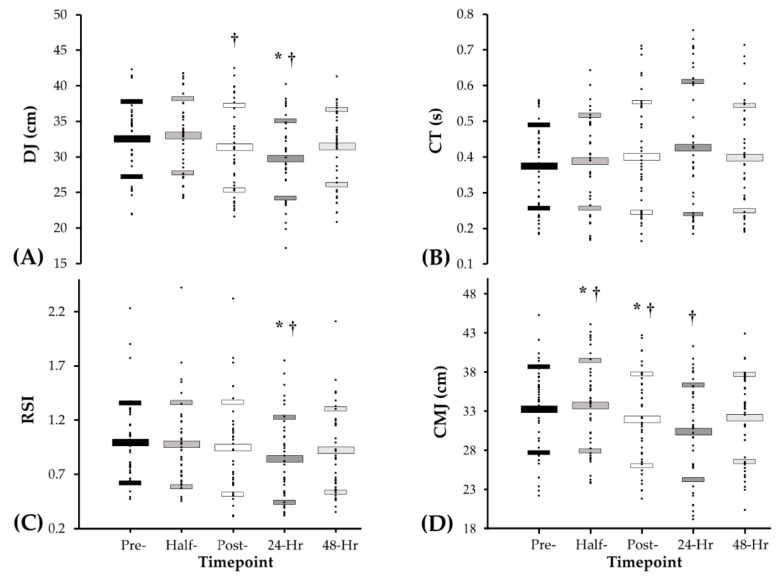
Values at pre-match, half-time, full-time, 24 h post-match and 48 h post-match for (**A**) drop jump height (cm), (**B**) drop jump contact time (s), (**C**) reactive strength index and (**D**) countermovement jump height (cm) (Data are presented as mean ± SD from left to right: pre-match, half-time, full-time, 24 h post-match and 48 h post-match) (* *p* < 0.05 vs. pre match; † *p* < 0.05 vs. half-time).

**Figure 3 sports-08-00166-f003:**
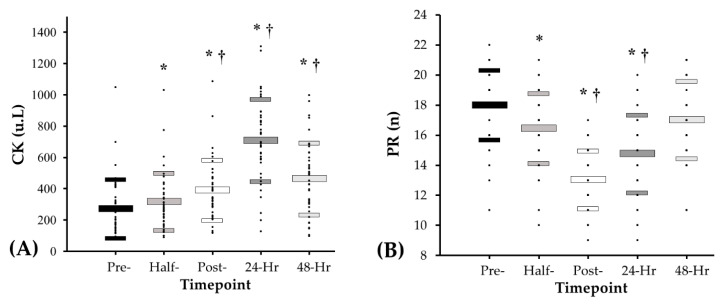
Values pre-match, half-time, full-time, 24 h post-match and 48 h post-match for (**A**) creatine kinase and (**B**) perceptual responses (Data are presented as mean ± SD from left to right: pre-match, half-time, full-time, 24 h post-match and 48 h post-match) (* *p* < 0.05 vs. pre match; † *p* < 0.05 vs. half-time).

**Table 1 sports-08-00166-t001:** Global Positioning System (GPS) metric classification using STATSports software.

Workload Metric	1st Half	2nd Half	Full Game
Max Speed (m·s^−1^)	8.2 ± 0.7	7.9 ± 0.5	8.1 ± 0.6
Total Distance (m)	3834.7 ± 654.0	3300.1 ± 714.2	7134.7 ± 1194.9
Total Accelerations (≥ 0.5 m·s·s^−1^) (*n*)	20.7 ± 6.7	15.4 ± 5.5	37.2 ± 11.4
Total Sprints (n) (≥5.5 m·s^−1^)	17.8 ± 8.3	12.8 ± 5.9	30.5 ± 11.9
Total Impacts (≥2g in a 0.1s) (*n*)	85.4 ± 81.3	47.6 ± 36.2	133.5 ± 122.4
Total Sprint Distance (≥5.5 m·s^−1^) (m)	426.4 ± 137.7	315.5 ± 119.7	742.0 ± 229.9
Average Heart Rate (beats m^−1^)	165.7 ± 9.2	170.2 ± 7.8	167.9 ± 8.3

Data presented as mean ± SD, 1st = First half of game, 2nd = second half of game.

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
