# Peer review of "Gaelic Football Match-Play: Performance Attenuation and Timeline of Recovery"

_sports, 2020, doi:10.3390/sports8120166_

Round 1

Reviewer 1 Report

From this point of view, although the study is carried out for a sport that is not widespread outside the UK, it is current. I have a few comments on the study that will need to be added. I did not find in the text the results of the evaluation of normality - Shapiro-Wilk test. The correlations evaluating motor variables, although statistically significant, determine these variables from about 30%. What are the other variables that determine the correlation relationships. The course of recovery depends on the type of training (interval - continuous), the period (beginning or end of the race period and the activities that players perform in the recovery phase. This information will need to be added to the text. are they very sensitive to current fatigue? Similarly, I lack the limits of the study.

Reviewer 2 Report

Dear Authors

You have written an interesting research paper. However, there are some things to address.

Sample : Were they from the same club? report

2.2.Anthropometrics and Body Composition

Who carried out the measurements? Where they ISAK certified? Report

Report reliability of anthropometric measurements.

2.3.Sampling of Plasma Creatine Kinase (CK

Where were the samples taken (in the lab – or in the club)? report

When were the samples analysed -  Immediately after being taken or later in the lab (If later – how were they stored and what was the time until tested)? Report

2.4.Perceptual Responses(PR)

‘’in private’’ did they take the questioners home?

2.5.Countermovement Jump(CMJ)and Drop Jump (DJ)

What were the instructions in the DJ – which leg starts – is put forward first? Report

Results:

Reactive Strength Index (RSI) – this the first time you mention that you calculated the RSI – add this to methods section and how it is calculated (add references)

There is no limitations of the studies paragraph (it was a training match, so there is a possibility the players didn’t go all out,..) – ADD

There is no real practical application of your findings for coaches – what does your study bring to them? How can they adapt the training, or what is your suggestion for recovery in the 24 - 48 hours after a match? ADD

Kind regards

Reviewer 3 Report

The article has a good quality, in the attached document for the authors I indicate some suggestions.

Reviewer 4 Report

The present study investigated changes in markers of fatigue and performance attenuation during and following competitive senior club level Gaelic football.

In general, the study is interesting and its redaction is clear and logical.

Abstract:

L10-11: indicate in the aim that it’s acute effect, only one match.

L13: please, delete abbreviations.

L21: please, define “max”

L24-25: “Furthermore, monitoring…” It is future perspective more than conclusions.

Introduction

L31 Please, add a reference.

L34 Please, add a reference.

L57 Please, add abbreviations of CMJ and DJ. It is the first time.

L64 Please, add abbreviations of PR. It is the first time.

Material and methods

L98 In my opinion, it would be more interesting include % of fat mass estimated since skinfolds.

Results

Table 1 In my opinion, it would be more interesting include data from first and second half of game, especially because then there are differences between half-time and others points.

L72 Sometimes you use “full-time” and in the figures you use “post-timepoint”. Please, unify.

L208 Please, delete #

 Figures Please, I suggest re-organize the figures. Why have you chosen this organization? For example, jump could be included in the same figure.

Discussion

L227 The objective is defined differently along the text. Please, review.

L245 Three studies are not numerous other studies. Please, re-write.

L266 For me, the most important point is the decrease in CMJ at 24.h post-match.

L271 Please, review.

Limitations? For example, the sample is not enough large. Also, you should mention the population. Amateur players? Review along the text.

Round 2

Reviewer 2 Report

Dear Authors

Thank you for addressing the raised questions. After reading your responses, the paper is, in my opinion, ready for publication.

Kind regards and keep up the good work